# Construction of Tissue-Engineered Bladder Scaffolds with Composite Biomaterials

**DOI:** 10.3390/polym14132654

**Published:** 2022-06-29

**Authors:** Wenjiao Li, Na Qi, Tingting Guo, Chao Wang, Ziwei Huang, Zhouyuan Du, Dingwen Xu, Yin Zhao, Hong Tian

**Affiliations:** 1Department of Medical Genetics, Tongji Medical College, Huazhong University of Science & Technology, Wuhan 430030, China; liwenjiao1990@yeah.net (W.L.); na245587796@163.com (N.Q.); 18739903963@163.com (T.G.); 2Hepatic Surgery Center, Institute of Hepato-Pancreato-Biliary Surgery, Department of Surgery, Tongji Hospital, Huazhong University of Science and Technology, Wuhan 430030, China; wang_chao0220@163.com; 3Department of Breast and Thyroid Surgery, Union Hospital, Tongji Medical College, Huazhong University of Science & Technology, No. 1277 Jiefang Avenue, Wuhan 430022, China; huangziwei_2008@163.com; 4Department of Gastrointestinal Surgery, Union Hospital, Tongji Medical College, Huazhong University of Science & Technology, No. 1277 Jiefang Avenue, Wuhan 430022, China; duzhouyuan@gmail.com; 5Department of Ophthalmology, Tongji Hospital, Huazhong University of Science & Technology, Wuhan 430030, China; dominicxu21@outlook.com

**Keywords:** bladder acellular matrix (BAM), poly lactic-co-glycolic acid (PLGA), Chitosan (CS), sodium alginate (SA), adipose-derived stem cells (ADSCs)

## Abstract

Various congenital and acquired urinary system abnormalities can cause structural damage to patients’ bladders. This study aimed to construct and evaluate a novel surgical patch encapsulated with adipose-derived stem cells (ADSCs) for bladder tissue regeneration. The surgical patch consists of multiple biomaterials, including bladder acellular matrix (BAM), collagen type I from rat tail, microparticle emulsion cross-linking polylactic-co-glycolic acid (PLGA)-chitosan (CS) with PLGA-sodium alginate (SA), and growth factors. ADSCs were seeded on the surgical patch. Approximately 50% of the bladder was excised and replaced with a surgical patch. Histological, immunohistochemical and urodynamic analyses were performed at the 2nd, 4th, and 8th weeks after surgery, respectively. The PLGA-CS, PLGA-SA or surgical patch showed no cytotoxicity to ADSCs. PLGA-CS cross-linked with PLGA-SA at a ratio of 5:5 exhibited a loose microporous structure and was chosen as the candidate for ADSC seeding. We conducted bladder repair surgery in rats using the patch, successfully presenting urothelium layers, muscle bundles, and vessel regeneration and replacing 50% of the rat’s natural bladder in vivo. Experiments through qualitative and quantitative evaluation demonstrate the application potential of the composite biomaterials in promoting the repair and reconstruction of bladder tissue.

## 1. Introduction

Cancer, inflammation, infection, trauma, spinal cord injury, and various congenital diseases, such as bladder eversion, can cause bladder dysfunction. In severe cases, the patient’s bladder fails, which requires radical cystectomy [1]. For many years, gastrointestinal segments have been used as replacement materials in bladder repair surgery. However, this approach results in complications such as malignant tumours, electrolyte imbalances, and urinary calculi [2]. Currently, tissue engineering alleviates the problem by repairing the damaged bladder using biodegradable scaffold material [3].

The bladder is a hollow organ, and its tube cavity wall is composed of the adventitia layer (visceral side), muscle bundles, blood vessels, nerve and bladder urothelial layer (the luminal side) from outside to inside. Owing to its complex structure, it is difficult to induce the desired differentiation into the above structure. Scaffolds, proliferative cells, and biosignalling are three main elements for the construction of tissue-engineered bladders. In the field of bladder tissue engineering, scaffolds play a role by simulating the natural microenvironment. It controls cell attachment, proliferation, and tissue regeneration [4].

Poly lactic-co-glycolic acid (PLGA) is extensively used in medical engineering because it possesses great characteristics, including biodegradability, biocompatibility, and nontoxicity. On account of its controllable degradation rate and microstructure, it can easily form a 3D structure. Nevertheless, studies show that PLGA can cause defective cell adherence and urinary bladder calculi, which is unfavourable for bladder reconstruction [5,6]. Chitosan (CS), a natural polysaccharide chitin, has many biological functions, such as biodegradability, antimicrobial activity, low toxicity, and immunogenicity [7]. It is widely used in antibacterial agents and artificial tissue materials. Sodium alginate (SA) is a natural polysaccharide whose usage is popular in medicine due to its stability, solubility, and viscosity [8]. As natural materials, CS and SA lack the ideal mechanical properties, but CS and SA have positive and negative charges, respectively, under suitable conditions, which can be beneficial to the formation of 3D structures together with PLGA [9].

Chitosan (CS) is composed of β (1–4) glycosidic bond-linked D-glucoamine and N-acetyl-D-glucosamine [10]. The particle size of chitosan nanoparticles is related to the molecular weight and deacetylation degree of chitosan, especially the molecular weight. The particle size of low molecular weight chitosan nanoparticles is approximately 10–20 nm, that of medium molecular weight chitosan nanoparticles is approximately 200 nm, and that of high molecular weight chitosan nanoparticles is more than 1 µm.

Ionically cross-linked beads of sodium alginate (SA) and methylcellulose (MC) were prepared as semi-interpenetrating polymer networks (semi-IPNs) in the size range of 1.97 ± 0.09−1.22 ± 0.13 mm by cross-linking with FeCl_3_ [11].

Polyester PLGA is a mutual blend of polylactic acid (PLA) and polyglycolic acid (PGA). The particle diameter of the PLGA composite was mostly between 100 nm and 1 µm. Magnetic nanocomposite biomaterial was synthesised by PLGA and magnetite for bone engineering, in which FeCl_2_/FeCl_3_ is bonded to a nanocomposite scaffold with a pore diameter of approximately 864 nm [12]. The nanofibre composition comprised 1 wt.% graphene oxide and 10 wt.% silk fibroin in PLGA and displayed a fibre diameter of 130 nm [13].

In the research of tissue engineering materials, the particle size of composite materials is mostly between 100 nm and 1 µm, and micro/nano is often used to express the particle size of composite materials [14,15].

A bladder acellular matrix (BAM) is commonly used as a scaffold material in tissue-engineered bladders because it has a 3D structure and contains essential growth factors, collagen, and so on [16]. Unfortunately, the shrinkage and scar formation of the graft in the experiment proved that BAM did not provide a suitable microstructure for cell proliferation and differentiation [17].

Collagen type I, an excellent substrate for cell culture, can be used to promote the adhesion of cells cultured in vitro, as well as to prepare 3D collagen gels. Collagen type I has high plasticity, good biocompatibility, low antigenicity, high biodegradability, and cell growth potential. However, the extensive use of collagen type I is hampered by poor mechanical strength [18].

When using a single type of material, it is difficult to overcome its defects. As a result, previous studies are not satisfactory in terms of clinical needs [19]. To compensate for the autogenous defects of a single type material, our approach involves the use of carefully chosen composite biomaterials.

With the above composite materials, we created a patch that was used to construct a better tissue-engineered bladder. We performed bladder reconstruction surgery using the patch, replacing 50% of the rat’s natural bladder in vivo, and successfully presented urothelium layers, muscle bundles, and vessels. Histological and immunohistochemical analyses and bladder capacity maximum (BCM) tests were conducted at the 2nd, 4th, and 8th weeks after surgery. In a statistical sense, we proved that our tissue-engineered bladder using the composite biomaterials as scaffolds could function normally. Our experiments demonstrated the application potential of our composite biomaterial in bladder repair and reconstruction.

## 2. Materials and Methods

### 2.1. Animals

Animal feeding and experiments were authorised by the Institutional Animal Research Committee of Huazhong University of Science and Technology (HUST) and in conformity with the Guide for the Care and Use of Laboratory Animals (National Institute of Health, Bethesda, MD, USA). The study was approved by the Animal Ethics Committee of HUST (HUST201712031).

### 2.2. Methods

#### 2.2.1. Adipose-Derived Stem Cells (ADSCs) Culture

Firstly, 3–8 g of inguinal adipose tissue was extracted from Sprague-Dawley (SD) rats, weighing 150–200 g, reared in the animal laboratory of the Huazhong University of Science & Technology (HUST). The adipose tissue was dissected, minced into small pieces, and digested in 0.1% collagenase type I (SCR103, Sigma-Aldrich, Shanghai, China) solution for 45 min at 37 °C. The digested adipose tissue was filtered through a 200 µm nylon mesh, and single ADSCs suspensions were cultured in high-glucose Dulbecco’s modified Eagle medium (DMEM, Thermo Fisher Scientific, Shanghai, China) supplemented with 10% fetal bovine serum (FBS; 10100147, Thermo Fisher) and 100 units/mL each of penicillin and streptomycin (ab287912, Abcam, Shanghai, China) in a plastic culture flask (37 °C, 5% CO_2_). The medium was replaced every 2 days. When the cells reached 90% confluence, they were sub-cultured with a cell-to-medium ratio of 1:3.

#### 2.2.2. PLGA-CS and PLGA-SA Cross-Linked Microparticle Emulsion Preparation

Chitosan (CS; 740179, Sigma-Aldrich, Shanghai, China) with a concentration of 2 mg/mL was dissolved in 0.2% acetic acid (64-19-7, Aladdin, Shanghai, China) solution (*v*/*v*). Sodium alginate (SA; A1112, Sigma-Aldrich, Shanghai, China) with a concentration of 2 mg/mL was dissolved in ddH_2_O solution. Then, 50 mL each of CS and SA solution were added into 20 mL of 10 mg/mL Poly (vinyl alcohol) (PVA; P139533, Aladdin, Shanghai, China) solution, respectively. The two solutions were stirred and ultrasonic to form CS colostrum and SA colostrum, respectively. PLGA (PLA/PGA = 75/25, viscosity = 0.3 dL/g, Shandong Institute of Medical Equipment, Jinan, Shandong, China) with a concentration of 10 mg/mL was dissolved in dichloromethane (75-09-2, Aladdin, Shanghai, China) solution. Then, 50 mL of 10 mg/mL PLGA solution was injected into CS colostrum, and the other 50 mL of the 10 mg/mL PLGA solution was injected into SA colostrum to create PLGA-CS and PLGA-SA microparticle emulsions. Both microparticle emulsions were washed using deionised water, sterilised with 0.1% peracetic acid, then washed again using deionised water, and finally desiccated using a 24-h lyophilisation cycle. The size and potential of both emulsions were estimated using the ZetaPALS dynamic light scattering system(ZetaPALS, Brookhaven, NY, USA). Positively charged PLGA-CS were cross-linked with negatively charged PLGA-SA in ratios of 3:7, 7:3, and 5:5 to form gel materials. Then the gel materials were coated with 20-nm platinum; its aperture was evaluated using scanning electron microscopy (SEM) (Nova NanonSEM 450, Hillsboro, OR, USA).

The toxicity of PLGA-CS and PLGA-SA were tested in passage 3 ADSCs. The cells were seeded with a density of 1 × 10^3^ cells/well in 96-well plates. PLGA-CS with 0.1, 0.2, 0.3, 0.4, and 0.5% as the five different concentrations, and PLGA-SA with 0.2, 0.4, 0.6, 0.8, and 1.0% as the five concentrations were added to each well of the 96-well plate, respectively. The CCK8 (BS350A, Biosharp Life Science, Beijing, China) assays were carried out according to the CCK8 kit manufacturer’s instructions. The CCK8 tests were conducted for 5 consecutive days to monitor cell growth.

Thermogravimetric analysis (TG) and Fourier Transform Infrared Spectroscopy (FTIR) were used to evaluate the compatibility of the cross-linked composites.

The PLGA-CS were cross-linked with PLGA-SA in ratios of 5:5, and then the gel microparticle sample was to be handed over to the Huazhong University of Science & Technology Analytical & Testing Center for testing. The instrument model was Pyris1 TGA (PerkinElmer Instruments, Billerica, MA, USA), and a nitrogen atmosphere was chosen.

The PLGA-CS were cross-linked with PLGA-SA in ratios of 5:5 and then freeze-dried; the powdered sample was to be handed over to the Huazhong University of Science & Technology Analytical & Testing Center for testing. The instrument model was a Nicolet iS50R (Thermo Scientific company, Waltham, MA, USA). The wavelengths ranged from 4000 to 500 cm^−1^ with a resolution of 4 cm^−1^.

#### 2.2.3. Bladder Acellular Matrix (BAM) Preparation

Bladders obtained from 1-year-old pigs were rinsed under running water. The bladder’s submucosa membrane was isolated and washed in distilled water at 4 °C and then placed in an oscillator for 2 days. The submucosa was then incubated, first with 0.5% Triton X-100 (85111, Thermo Fisher Scientific, Shanghai, China) at 4 °C, oscillating for 2 days, and then with 0.05% ammonium hydroxide (1336-21-6, Aladdin, Shanghai, China) for another 2 days to thoroughly remove cellular debris. The obtained BAM was washed with distilled water, then sterilised with 0.5% peracetic acid, washed again with distilled water, and lastly stored in phosphate-buffered saline at 4 °C. Before the experiment, the BAM was rehydrated with PBS containing PDGF-BB (platelet-derived growth factor subunit B, 1μg/mL; PDB-H5127, ACROBiosystems, Newark, DE, USA), VEGF (Vascular endothelial growth factor A, 1μg/mL; VE0-H5212, ACROBiosystems, Newark, DE, USA) and TGF (Transforming growth factor-beta 1, 1μg/mL; TG1-H4212, ACROBiosystems, Newark, DE, USA), incubated at 37 °C overnight.

#### 2.2.4. Preparation of Surgical Patch

The prefabricated hydrated patch was produced through the following procedures: 200 μL microparticle emulsion was added, cross-linking PLGA-CS with PLGA-SA (0.1 g/mL, pre-suspended in 10× DMEM), 140 μL FBS and 60 μL growth factor (20 μL 1μg/mL PDGF-BB), 20 μL 1μg/mL VEGF and 20 μL 1μg/mL TGF to 0.6 mL sterile collagen type I (C0130, Sigma-Aldrich, Shanghai, China, 3.65 mg/mL), and neutralised with 40μL 1 M NaOH (1310-73-2, Aladdin, Shanghai, China). The materials were kept on ice. The mixed solution was instantly placed in an incubator (37 °C, 5% CO_2_) for 2 h to jellify the prefabricated hydrated patch.

To form a surgical patch, the following materials were deposited layer by layer from top to bottom: tissue paper, nylon meshes, hydrated patch, rehydrated BAM, another layer of nylon meshes and another layer of tissue paper, under sterile conditions. Then, static compressive stress was applied on top of the gels for 1–2 min to extrude the moisture.

In order to evaluate the cell differentiation induced by composite materials, the jelly-like patches were cut into 4 cm × 4 cm pieces and distributed into 6-well plates. Passage 3 ADSCs (5 × 10^5^ cells/well) were seeded into the 6-well plates with the patches at the bottom, and BAM at the bottom was used as a control group. The 3: 1 DMEM and keratinocyte serum-free medium (KSFM; 10744019, Thermo Fisher, Waltham, MA, USA) was a special medium in which DMEM contains 10% FBS and growth factor (1 μg/mL PDGF-BB, 1 μg/mL VEGF, 1 μg/mL TGF). The medium was replaced every 3 days. On the 7th day, the cells were collected, and the expressions of the urethral epithelial cells and smooth muscle cells were detected by Cytokeratin 13 (CK13; sc-53265, Santa Cruz, CA, USA, 1:100), Uroplakin Ia (UP-1α; ab185970, Abcam, Cambridge, MA, USA 1:100), myosin (ab37484, Abcam, Cambridge, MA, USA, 1:50), and Desmin (sc-23879, Santa Cruz, CA, USA, 1:100), respectively.

To detect the growth of cells on the patch, the jelly-like patches were then cut into 2 cm × 2 cm pieces and distributed into 24-well plates. Passage 3 ADSCs (4 × 10^4^ cells/well) were seeded into the 24-well plates with the patches at the bottom and cultured in DMEM with 10% FBS, while the patches without seeded cells were simultaneously maintained as a control group in 24-well plates. The CCK-8 assays were conducted for 5 consecutive days to monitor the continuous proliferation of ADSCs on the patch.

The patch, with or without seeded ADSCs, was used as the surgical repair material after 24h of culture.

#### 2.2.5. Transplantation Experiment in Rats In Vivo

SD rats, each weighing from 200 to 250 g, were purchased from the Animal Center of HUST. The rats were randomly allocated into three groups: sham operation group, patch seeded cells used as the grafting test group, and patch without seeded cells used as the grafting control group. Each group consisted of 9 rats, and each test used 9 rats, respectively, from 3 different groups to compare each group’s cell growth. The retention capacity of the bladder was determined before the grafting operation, and the engineered bladder was surgically obtained on the 2nd, 4th, and 8th week after the operation.

Before the operation, the rats were injected with penicillin to reduce the risk of infection and then anaesthetised with chloral hydrate (302-17-0, Aladdin, Shanghai, China, 7%, 0.5 mL/100 g). Approximately 50% of the bladder was excised and replaced with the patch without seeded cells (control) or with the patch seeded cells (test group). The sham-operation group simply sewed up the excised bladder. Water-tight integrity was tested by filling with saline solution.

#### 2.2.6. Bladder Capacity Maximum (BCM) Test

The bladder capacity and urinary retention of each rat were evaluated using a multichannel physiological signal acquisition system, RM6240BD (CHENG YI, Chengdu, China), and a constant flow pump (B. Braun Melsungen AG, Melsungen, Germany). Under chloral hydrate anaesthesia, a modified epidural catheter was inserted into the rat bladder through the urethra and was connected to the pump. After emptying the urine, saline solution (7647-14-5, Aladdin, Shanghai, China, 0.85% NaCl) was infused into the bladder at 30 mL/h through the pump. When urine leakage first occurred, the bladder had reached its maximum capacity. The BCM test should be conducted for all rats during the 2nd, 4th, and 8th week after the surgical operation.

#### 2.2.7. Histological and Immunohistochemical Analyses of the Cell Growth in Patch

Regular hematoxylin (H9627, Sigma-Aldrich, Shanghai, China) and eosin (861006, Sigma-Aldrich, Shanghai, China,) (H&E) and Masson’s trichrome (1.00485, Sigma-Aldrich) staining were performed to analyse whether tissue structures such as urothelium layers, muscle bundles, and blood vessels were present in the newly-constructed bladder. Cytokeratin 13 (CK13; sc-53265, Santa Cruz, CA, USA, 1:100) antibodies were used to identify the reconstructed urothelial cells, while α-smooth muscle (SMA; sc-53142, Santa Cruz, CA, USA, 1:100) antibodies were used to identify the smooth muscle cells (SMCs) and blood vessels.

## 3. Results and Discussion

### 3.1. Characterisation of PLGA-Microparticles

The PLGA-CS microparticle has a diameter of 202.2 ± 13.7 nm and a zeta potential of 14.41 ± 2.37 mV; the PLGA-SA microparticle has a diameter of 143.4 ± 5.2 nm and a zeta potential of -24.72 ± 0.67 mV (Figure 1A,B). All three cross-linked materials using PLGA-CS and PLGA-SA emulsions in ratios of 3:7, 7:3, and 5:5 exhibited a loose microporous structure, with a pore diameter between 100 nm and 400 nm. Among the three materials, the one with a ratio of 5:5 was chosen as the candidate PLGA-microparticle for the next experiment since its pore size distribution is uniform (Figure 1C). The CCK-8 assays showed that PLGA-microparticles (CS and SA) had no cytotoxicity to ADSCs (Figure 1F,G).

A TG test showed that the polymer PLGA began to lose weight at about 150 °C, lost 50% at 288 °C, quickly lost weight between 250 and 350 °C, dropped to 1.48% at 350 °C, and only lost 1.23% at 600 °C. PLGA-CS cross-linked PLGA-SA to form two weightless platforms. The weight loss started at a low temperature of 31.77 °C, which was due to the intervention of polysaccharides CS and SA, which quickly reduced the initial weight loss temperature. The weight loss of PLGA-CS cross-linked PLGA-SA is about 50% at 55 °C, and it decreases to 12% at 66 °C, forming the first weight loss step. When the temperature is close to 200 °C, the second weightlessness step begins. The weight dropped to 0.4% at about 500 °C, then slightly increased to 0.5%, and maintained until the end of 600 °C, and the components nearly disappeared. With the addition of CS, the absorbance curve of PLGA was lifted as a whole, but after PLGA-CS cross-linked PLGA-SA, the absorbance moved back to the vicinity of the PLGA curve (Figure 1D).

At the wave number of 1624-1625, a new absorption peak appeared due to the intervention of CS, which was not found in PLGA. It is speculated that the absorption peak may be caused by NH2 of CS. At the wave number of 1756-1758, the absorbance of PLGA, PLGA-CS, and PLGA-CS cross-linked PLGA-SA are the lowest values of the curve absorbance, which are around 48%, 50%, and 28%, respectively (Figure 1E). Presumably, this is mainly due to −C=O stretch out and draw back. In short, the addition of CS and SA has little effect on the absorption peak of PLGA, which indicates that the three have good compatibility.

### 3.2. Prepared Bladder Acellular Matrix (BAM) and Surgical Patch

The bladder acellular matrix (BAM), a native constituent of the bladder, contains some secretion factors produced by different cells in the bladder tissue. These factors can be preserved in the decellularisation process. H&E and Masson’s trichrome staining showed that our BAM had almost no residual host porcine cells, as shown in Figure 2A,B. The CCK-8 assays showed that the surgical patch had no cytotoxicity to ADSCs (Figure 2C).

The jelly-like patch, as a scaffold, is a translucent thin film (Figure 2D(a)). When examined 2 weeks after surgery, the graft patch was white and opaque, showing no signs of shrinkage or calculi in vivo (Figure 2D(b)). Recent studies have shown that in the large animal bladder model constructed by the BAM scaffold alone, there are some problems, such as fibrosis, shrinkage of the graft, and insufficient bladder volume [20]. This composite material has the potential to be used in large animal models and clinics.

### 3.3. Effect of Surgical Patches Containing PLGA Microparticles on ADSCs

The ADSC differentiation efficiency was evaluated by immunofluorescence and Western blotting in vitro. Here, we used UP-1α and CK13 as urothelial epithelial cell markers and myosin and desmin as smooth muscle cell markers. As shown in Figure 2E,G, the surgical patch induced ADSCs to express more UP-1α and CK13. Similar to the trend shown in Figure 2F,H more myosin and desmin were expressed within the surgical patch induction. These results indicated that patches containing PLGA-microparticles could efficiently induce ADSC differentiation.

### 3.4. Histological and Immunohistochemical Analyses of Cell Growth in Patches

The cross-section of the suture patch shows the shape of the bladder wall. Appendix A shows the cell growth by haematoxylin and eosin (H&E) staining. By the 2nd week after surgery, there were a large number of urothelial cells that immigrated to the marginal zone in the control group to form urothelium layers, and the number of cells that immigrated in the central area was relatively less than that in the marginal zone (Appendix A). By the 4th week after surgery, compared with the control group, there were more cells in the marginal area and central area of the test group (Appendix A). By the 8th week after surgery, the number of cells in the central area was reduced in both the control group and the test group (Appendix A). Implanted cells on scaffolds are a strategy of tissue engineering [21]. Although the results of our test group with seeded cells were better than those of our control group without seeded cells, our control group also showed obvious tissue differentiation.

Masson’s trichrome staining was performed to evaluate cell proliferation. As shown in Figure 3A, by the 2nd week, the control group (Figure 3A(a)) and the test group (Figure 3A(d)) revealed urothelium layers (indicated by arrows) in the marginal zone. By the 8th week, a large number of urothelial cells were arranged in an orderly manner to form urothelium layers in both the test group and the control group (Figure 3A(c,f)).

Cytokeratin 13 antibodies were used to identify urothelial cells. From the 2nd to the 8th week, cytokeratin 13 was positive in the urothelium layers in both the test group and the control group (Figure 3B indicated by arrows). By the 8th week, the urothelium layers in the control group (Figure 3B(c)) and in the test group (Figure 3B(f)) were thicker than those in the sham operation group (Figure 3B(g)). This indicates that the urothelial cells were aligned in order and formed urothelium layers. This result is consistent with the H&E and Masson’s trichrome staining results shown in Figure 3A.

Masson’s trichrome staining was also used to evaluate the proliferation of smooth muscle cells (SMCs) in the central zone (Figure 4A). On the 2nd day, neither the control group (Figure 4A(a)) nor the test group (Figure 4A(d)) showed the appearance of detailed muscle bundles. Starting at the 4th week, SMCs gradually increased and formed muscle bundles in the test group (asterisks indicate muscle bundles). Until the 8th week, there were detailed muscle bundles present in both the control group(Figure 4A(c)) and the test group (Figure 4A(f)). The number of SMCs in the test group (Figure 4A(d–f)) was greater than that in the control group (Figure 4A(a–c)), and the muscle bundles in the test group(Figure 4A(f)) were thicker than those in the control group (Figure 4A(c)). However, the muscle bundle thickness of the control group and the test group is not as good as that of the sham operation group(Figure 4A(g)). Even though collagen and growth factors can be preserved in the decellularisation process [22], we think that growth factors will still be lost during decellularisation. To compensate for the loss of growth factors in BAM, we selectively added the growth factors PDGF-BB and TGF-β, which are beneficial to the formation of smooth muscle [23]. In previous studies, SMC growth was discontinuous, incomplete, and often formed disordered muscle fibres [24,25].

The α-smooth muscle antibodies were used to identify the SMCs (Figure 4B). From the 2nd to the 8th week, the muscle bundle gradually increased in both the control group and test group; by the 8th week, there were clear muscle bundles that were α-SMA-positive, as indicated by the asterisks in Figure 4B(f). From the 4th to the 8th week, there were more SMCs in the test group(Figure 4B(e,f)) than in the control group (Figure 4B(b,c)). The muscle bundle presented in the test group (Figure 4B(f)) was thicker than that in the control group (Figure 4B(c)).

The α-smooth muscle antibodies also revealed the reconstruction of the vasculature (Figure 5). By the 2nd week, blood vessels began to form in both the control group (Figure 5a) and the test group (Figure 5d). From the 4th week to the 8th week, the number of blood vessels in the test group gradually increased (Figure 5e,f), and the number of blood vessels in the control group(Figure 5c) increased obviously until the 8th week. However, the blood vessels in the two groups were obviously not as thick as those in the sham-operation group(Figure 5g). Currently, the vascularisation of the reconstructed tissues is achieved mainly via capillaries. The major blood vessels that formed are less than required for an entire tissue or organ, and therefore, the blood supply is often insufficient in previous studies [26,27]. To enhance the blood supply, the growth factor VEGF, which is beneficial to the formation of vessels [28], was added.

### 3.5. Bladder Capacity of the Reconstructed Bladder

Figure 6 displays the results of the BCM test of the reconstructed bladder. BCM is measured by the maximum volume of saline solution pumped to the bladder after emptying the urine. The mean BCM was 1.23 ± 0.1 mL before the graft operation. The BCM of the reconstructed bladder using the test group increased to 1.87 ± 0.10 mL by the 2nd week and 1.64 ± 0.30 mL by the 4th week and remained at the same level during the remaining 4 weeks (1.70 ± 0.41 mL). The data were significantly higher than the preoperative average. In the control group, the BCM increased to 1.78 ± 0.14 mL by the 2nd week, then decreased drastically by the 4th week (0.88 ± 0.20 mL) and finally reached the normal level (1.37 ± 0.76 mL) by the 8th week. Overall, the retention capacity of the reconstructed bladder was not significantly different from that of a normal bladder.

Except for the 4th week, the BCM of the engineering bladder in the BAM control group was lower than the average value before the operation (1.23 ± 0.1 mL), and the BCM detected at other time points exceeded the average value before the operation, suggesting that the engineered bladder had a normal function, and its structure was not different from that of the normal smooth bladder with no obvious shrinkage or scarring.

## 4. Conclusions

We constructed a 3D composite material as a scaffold for bladder regeneration surgery patches. The patch consists of BAM, collagen type I, growth factor, and microparticle emulsion cross-linking PLGA-CS with PLGA-SA. In vitro experiments proved that the composite materials have no cytotoxicity to ADSCs; in vivo, our graft surgery demonstrated that the engineered bladder we repaired formed urothelium layers, muscle bundles, and vessels. The BCM test showed that the neobladder could function normally, containing the same volume of urine as the normal bladder. Our composite materials have made up for the biological defects of synthetic materials and the mechanical defects of natural materials.

## Figures and Tables

**Figure 1 polymers-14-02654-f001:**
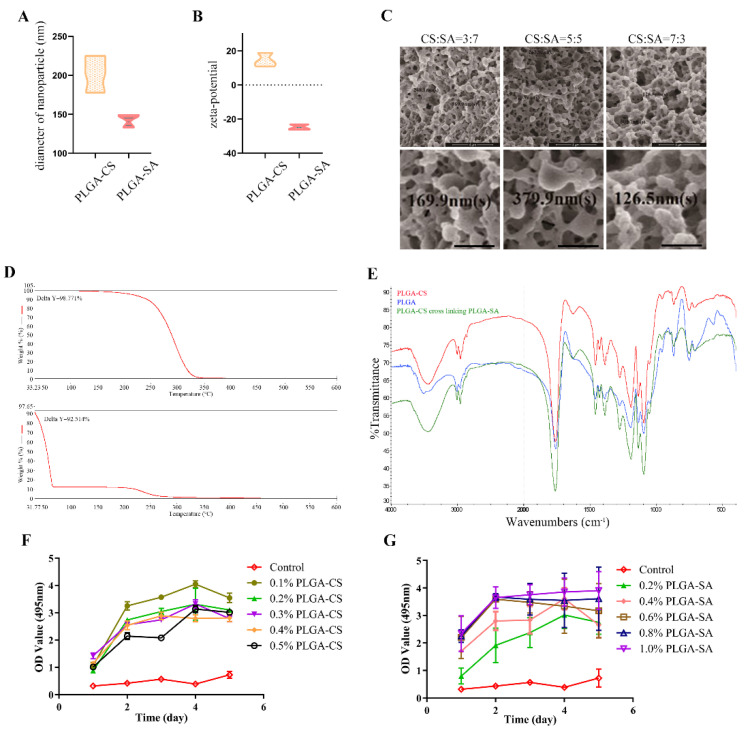
Characterisation of PLGA-microparticles. (**A**) Diameter of microparticles. (**B**) Zeta-potential of microparticles. (**C**) Scanning electron microscopy images of PLGA-CS and PLGA-SA cross-linked microparticles. The black number indicates particles diameter of the cross-linked microparticles. All three cross-linked materials exhibited a loose microporous structure, with a pore diameter between 100 nm and 400 nm. (**D**) TGA curve from PLGA (top), PLGA-CS cross-linking PLGA-SA (bottom) microparticles at temperatures between 33 and 600 °C. (**E**) FTIR spectra of PLGA-CS (red), PLGA (blue), PLGA-CS cross-linking PLGA-SA (green). (**F**,**G**) CCK8 assay ADSCs proliferation on different concentrations of PLGA-CS and PLGA-SA on five consecutive days.

**Figure 2 polymers-14-02654-f002:**
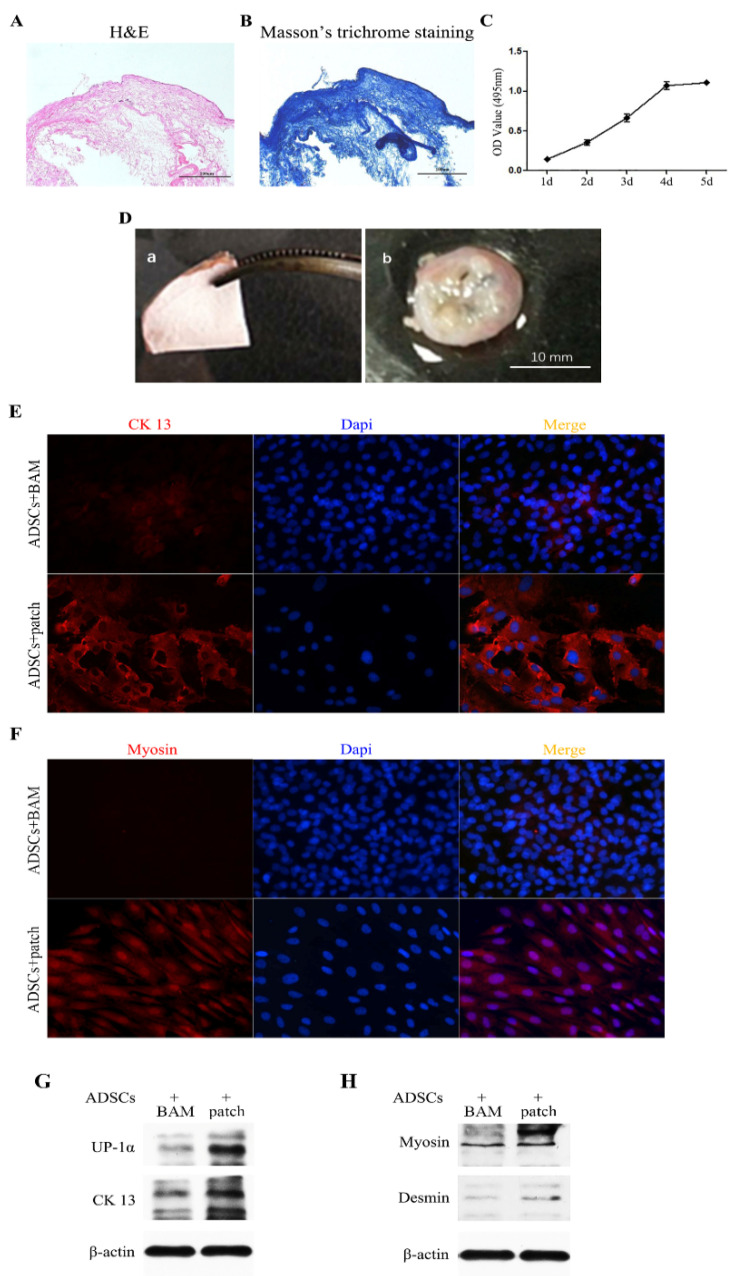
The surgical patch preparation and effect of ADSCs. (**A**,**B**) H&E and Masson’s trichrome staining showed almost no residual host-porcine cells. (**C**) CCK8 assay ADSCs proliferation on the surgical patch in five consecutive days. (**D**) Morphological features of the jelly-like patch. (**left**) Before graft patch. (**right**) Patch in the rat bladder at 2nd week after surgery. (**E**) CK13 immunostaining of ADSCs encapsuled by surgical patch or BAM. (**F**) Myosin immunostaining of ADSCs encapsuled by surgical patch or BAM. (**G**) Western blot results indicated the expression of UP-1α and CK13 in ADSCs encapsuled by surgical patch or BAM. (**H**) Western blot results indicated the expression of myosin and desmin in ADSCs encapsuled by surgical patch or BAM.

**Figure 3 polymers-14-02654-f003:**
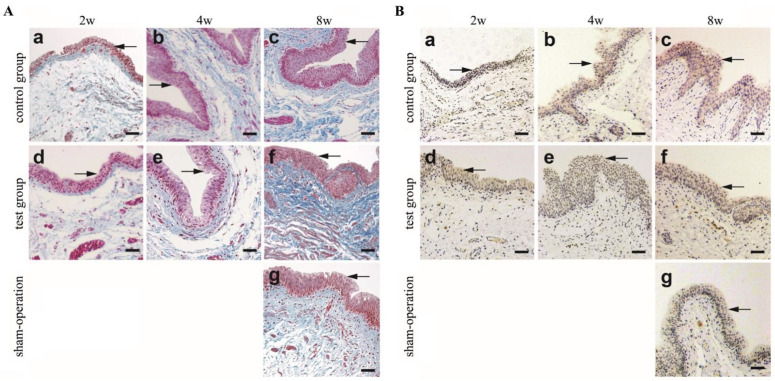
Masson’s trichrome and immunohistochemical staining with CK 13 to identify urothelial cells. (**A**). The results showed that Masson’s trichrome stained epithelial cells gradually increased with time (2w to 8w). At 2nd week, the control group (a) and the test group (d) revealed that urothelium layers in marginal zone. At 8th week, a large number of urothelial cells are arranged orderly to form urothelium layers in control group (c) and test group (f). The epithelial cell density of control group, test group and sham-operation group (g) was similar by Masson`s trichrome staining (**left**). (**B**). The results showed that CK13 stained epithelial cells gradually increased with time (2w to 8w). At 2nd, 4th, and 8th week after surgery, the urothelial cells growth in the control group (a–c) and test group (d–f). At 8th week, the urothelium layers in control group (c) and in test group (f) was thicker than that in the sham-operation group (g) by Immunohistochemical staining with CK 13 (**right**). Urothelial cells are indicated by arrows. (Scale bar = 50 µm).

**Figure 4 polymers-14-02654-f004:**
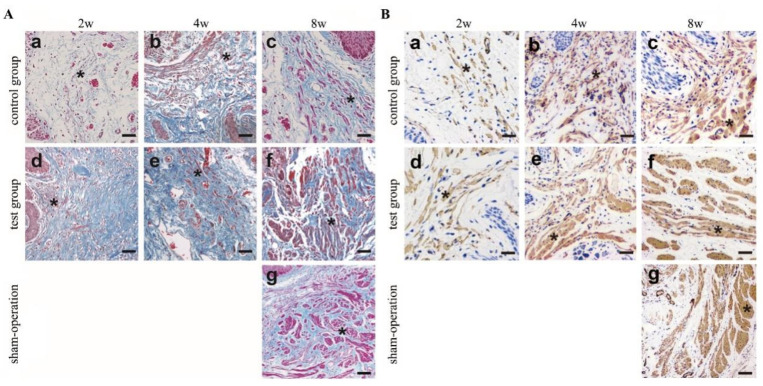
Masson’s trichrome and immunohistochemical staining with α-SMA to identify the SMCs. (**A**).The results showed that Masson’s trichrome stained. The number of SMCs gradually increased with time (2w to 8w). At 2nd, 4th, and 8th week after surgery, the SMCs proliferation of the central zone in the control group (a–c) and test group (d–f). Until 8th week, there were detailed muscle bundles present both in control group (**c**) and test group (f). The number of SMCs in test group (f) were more than that in control group (c) and the muscle bundles in test group (f) were thicker than that in control group (c). However, the muscle bundle thickness of the control group and the test group is not as good as that of the sham operation group(Figure 4A(g)), by Masson`s trichrome staining (**left**). (**B**).The results of α-SMA staining once again confirmed the results of Masson’s trichrome staining to SMCs. From 2nd to 8th week after surgery, the SMCs gradually increased in both the control group (a–c) and test group (d–f). At 8th week, both the control group (c) and the test group (f) showed clear muscle bundles. The muscle bundle presented in test group (f) was thicker than that in control group (c), the muscle bundle thickness of the control group and the test group is not as good as that of the sham operation group(Figure 4B(g)) by immunohistochemical staining with α-SMA (**right**). Asterisks indicate SMCs or muscle bundles. (Scale bar = 50 µm).

**Figure 5 polymers-14-02654-f005:**
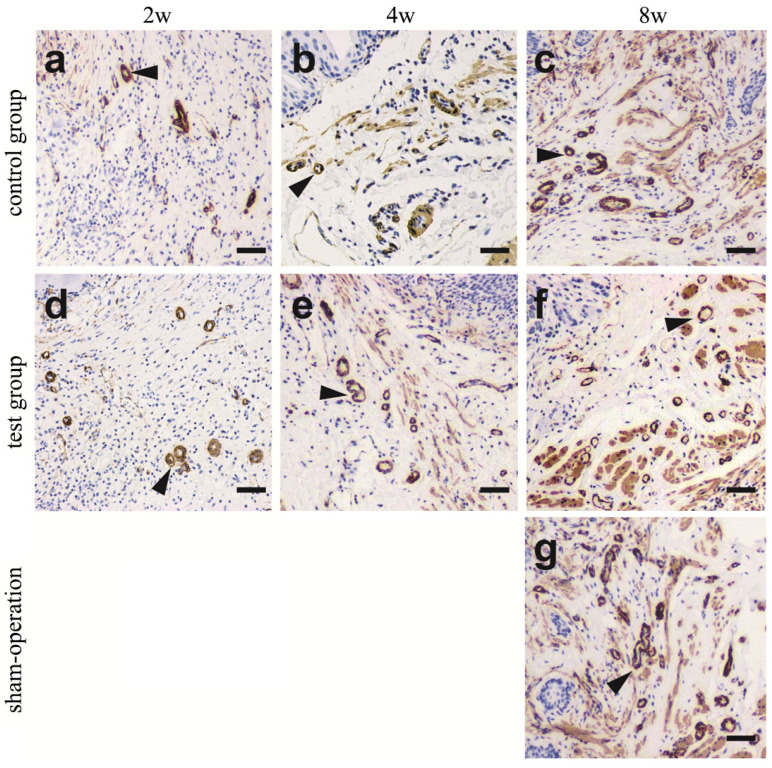
Immunohistochemical staining with α-SMA to trace the formation of blood vessels. At 2nd, 4th, and 8th week after surgery, the blood vessels regeneration in control group (**a**–**c**) and test group (**d**–**f**). With the passage of time (2w to 8w), the number of blood vessels gradually increased, especially in the test group(**d**–**f**). At the 8th week, the blood vessels in the control group (**c**) and the experimental group (**f**) were not as thick as those in the sham-operation group (**g**). Arrowheads indicate vessels. (Scale bar = 50 µm).

**Figure 6 polymers-14-02654-f006:**
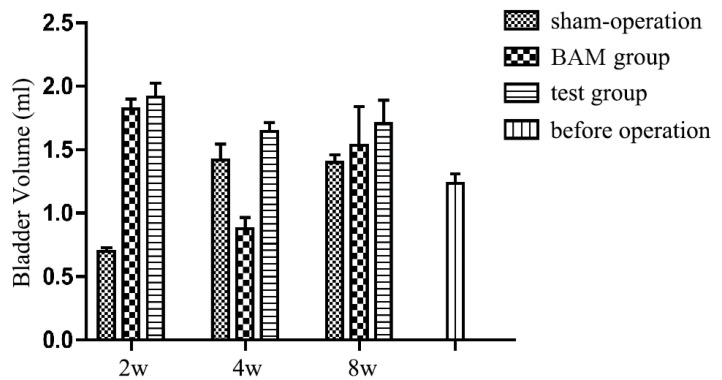
Quantity comparison of maximal bladder capacity in different groups at 2nd, 4th, 8th week after surgery. Maximum capacity was recorded when leakage first occurs.

## Data Availability

All the data can be available in Reference.

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
