# Peer review of "Construction of Tissue-Engineered Bladder Scaffolds with Composite Biomaterials"

_polymers, 2022, doi:10.3390/polym14132654_

Round 1

Reviewer 1 Report

Dear Authors and Editor,

I read the paper, that is interesting but some issues must be clarified:

1. Please define in the abstract PLGA.

2. In the Materials and Methods section, add all the reagents used.

3. Figure 1 needs magnification.

4. Please add a scheme for chapter 2.2.2.

5. Please check the English by a native speaker.

Author Response

  1. Please define in the abstract PLGA.

Thanks for the suggestion, we have revised the definition of PLGA in the abstract.

  1. In the Materials and Methods section, add all the reagents used.

Thanks for the suggestion, we add the reagents including all the antibodies information in the revised method section highlight in yellow.

  1. Figure 1 needs magnification.

Thanks for the suggestion, we reorganized figure 1 in the revised manuscript.

  1. Please add a scheme for chapter 2.2.2.

Thanks for the suggestion, we add a scheme for chapter 2.2.2 in the revised manuscript from line 125 to line 135.

  1. Please check the English by a native speaker.

Thanks for the suggestion, we invited a native speaker to check the grammar. Here is the certification in the attachment.

Reviewer 2 Report

The paper "Construction of tissue engineered bladder scaffold with composite biomaterials" aims to present a 3D composite material as a scaffold for bladder regeneration surgery patch, still the polymer composite structure is not investigated in detail, while the main results refer to the biocompatibility of the composite patch. Therefore, the first general recommendation regarding the manuscript in order to better fit the profile of Polymers Journal is a deeper investigation of the 3D polymer composite patch.

Introduction could be enriched with some physical-mechanical properties of individual (bio)polymers and similar composites, such as rheological behavior, flexibility, stretching resistance, adhesion, swelling-releasing properties, aiming to motivate the chosen composite system for the current study.

In Section 2 please define all the abbreviations at their first appearance in text, for example PDGF, VEGF, TGF;

L110: dL/g instead of dl/g;

3.1. L197-8: Generally, only the particles <100 nm are called nanoparticles, above 100nm they are microparticles; also, from Fig.1C the scale is not visible and no particulate shape can be observed, the polymer composite looking rather microfibrilar with micro/meso/macropores, therefore the description of the systems should be revised. The size of Fig.1C could be increased also.

For Figures 1D,E please define "Mock" and also use different colors for the OD lines;

The cross-linking of the polymer particles into composite systems should be more thoroughly investigated and discussed, for example by FTIR, Raman, rheology, thermogravimetry, etc. 

L223: Here, some shrinkage experiments for the composite systems at different temperatures and time intervals would be relevant.

Please revise the citation style of Polymers-MDPI: "In the text, reference numbers should be placed in square brackets [ ], and placed before the punctuation;".

Author Response

1, The paper "Construction of tissue engineered bladder scaffold with composite biomaterials" aims to present a 3D composite material as a scaffold for bladder regeneration surgery patch, still the polymer composite structure is not investigated in detail, while the main results refer to the biocompatibility of the composite patch. Therefore, the first general recommendation regarding the manuscript in order to better fit the profile of Polymers Journal is a deeper investigation of the 3D polymer composite patch.

Thank you so much for the comments. According to the suggestion, we examined the characteristic of the composite patch with FTIR and thermogravimetry. Here are the results in the below. We also revised the manuscript in the introduction and discuss section.

2, Introduction could be enriched with some physical-mechanical properties of individual (bio)polymers and similar composites, such as rheological behavior, flexibility, stretching resistance, adhesion, swelling-releasing properties, aiming to motivate the chosen composite system for the current study.

According to the suggestion, we revised the introduction section from line 63 to line 80.

3, In Section 2 please define all the abbreviations at their first appearance in text, for example PDGF, VEGF, TGF;

Sorry for the mistake, we re-defined all the abbreviations at the first appearance in the revised manuscript.

4, L110: dL/g instead of dl/g;

Sorry for the wrong typing, we revised it in the manuscript line xxx.

5, L197-8: Generally, only the particles <100 nm are called nanoparticles, above 100nm they are microparticles; also, from Fig.1C the scale is not visible and no particulate shape can be observed, the polymer composite looking rather microfibrilar with micro/meso/macropores, therefore the description of the systems should be revised. The size of Fig.1C could be increased also.

Thanks for the suggestions. We reorganized figure 1C with increased size, and revised the description of the systems.

6, For Figures 1D, E please define "Mock" and also use different colors for the OD lines;

Sorry for the misunderstanding, “Mock” means blank control without drug treatment. For simplicity, we changed Mock to Control in the revised figure 1.

7, The cross-linking of the polymer particles into composite systems should be more thoroughly investigated and discussed, for example by FTIR, Raman, rheology, thermogravimetry, etc.

Thanks for the suggestion, we examined the characteristic of the polymer particles with FTIR and thermogravimetry. Here are the results in the below.

8, L223: Here, some shrinkage experiments for the composite systems at different temperatures and time intervals would be relevant.

Thanks for the suggestion, we added a paragraph from line 364 to line 368, and explained the possibility of no obvious shrinkage with the experimental results.

9, Please revise the citation style of Polymers-MDPI: "In the text, reference numbers should be placed in square brackets [ ], and placed before the punctuation;".

We are sorry for the wrong citation style, we revised the style in the revised manuscript.

Round 2

Reviewer 1 Report

  1. Please add a scheme for chapter 2.2.2.

Author Response

Thanks for the suggestion, in revised chapter 2.2.2, we added a scheme from line 125 to line 160. 

Reviewer 2 Report

Dear Authors, thank your for revising the manuscript. Still, in the last version v2, the FTIR and TGA results do not appear, or any mention of them.

Firstly, i recommend the addition of the FTIR and TG analyses methods in sub-section 2.2.4. or in a new sub-section, while the results could be embedded in sub-section 3.1. Also, the FTIR figure could be enriched with the spectra of individual compounds, with the identification of main groups vibrations and specific bonds of the composite. Also, please add Wavenumber (cm-1) on the abscissa and narrow the interval to 500-4000 cm-1.

Regarding the TGA results, the derivative curve can be added below the mass loss curve in order to evidence clearer the thermal degradation of the composite.

L67,68: Please revise "nanoparticles" to "microparticles". 

Author Response

Thank you so much for the suggestion, we have added FTIR and TGA test analysis on the microparticles in figure 1. Also, we have added the method for FTIR and TGA test from line 150 to line 160.